# Mastering Your Dragons: Using Tabletop Role-Playing Games in Therapy

**DOI:** 10.3390/bs15040441

**Published:** 2025-03-31

**Authors:** Sherry R. Rosenblad, Tessa Wolford, Richard S. Brennan, Josh Darnell, Challen Mabry, Andrew Herrmann

**Affiliations:** 1Department of Clinical Counseling and Mental Health, Texas Tech University Health Sciences Center, 3601 4th St., Lubbock, TX 79430, USA; andrew.herrmann@ttuhsc.edu; 2Department of Counselor Education and Supervision, Walden University, 100 Washington Ave. S., Minneapolis, MN 55401, USA; tessa.wolford@crossroadscounseling.life; 3Department of Psychiatry, Texas Child Health Access Through Telemedicine, Texas Tech University Health Sciences Center, 1674 Pine St., Abilene, TX 79601, USA; richard.brennan@ttuhsc.edu; 4Department of Counselor Education, Bluefield University, 3000 College Dr., Bluefield, VA 24605, USA; jadarnell@bluefield.edu (J.D.); cmabry@bluefield.edu (C.M.)

**Keywords:** creative methods in psychotherapy, imagination in psychotherapy, tabletop role-play games, Dungeons & Dragons, therapeutic technique

## Abstract

Using Dungeons & Dragons (D&D) as a therapeutic tool is more feasible than previously thought. While role-playing games (RPGs) have existed for decades, their application in therapy can be challenging for those unfamiliar with tabletop gaming. This article explores the history of D&D, its therapeutic applications, and case studies of five individuals (ages 7–19) using RPGs addressing anxiety, depression, Dissociative Identity Disorder (DID), and nightmares. It also examines group therapy settings for individuals with Autism Spectrum Disorder (ASD), LGBTQ adolescents exploring identity, and women in residential treatment for substance use and trauma recovery. Each case study highlights both the successes and challenges of integrating RPGs into therapy, demonstrating how D&D fosters social skills, identity exploration, trauma recovery, and emotional growth. The findings suggest that RPGs are flexible and effective tools for addressing diverse mental health concerns by encouraging emotional exploration and personal development. While the results are promising, further research is needed to assess the long-term impact of RPGs in therapy and their broader clinical applications. Expanding this research could solidify D&D’s role as a valuable therapeutic intervention across various mental health settings.

## 1. Introduction

For over 50 years, the tabletop role-playing game Dungeons and Dragons (D&D) has been a genre-defining force in popular culture, inspiring movies, books, television, and games ([19]). Originally a niche hobby for creators Gary Gygax and Dave Arneson in the early 1970s, D&D has evolved into a cultural phenomenon, blending storytelling, roleplay, and strategy ([19]). Its narrative transference and role-play structure provide fertile ground for therapeutic potential in enhancing social skills, empathy, distress regulation, and problem-solving abilities ([1]; [4]; [7]; [8]; [12]; [20]; [18]; [25]). The continued use of D&D has led to adaptations and the creation of a gaming genre now referred to as Tabletop Roleplaying Games (TTRPGs; [11]). This article explores the history of D&D and other TTRPGs, provides an overview of gameplay, and examines how the game’s mechanics can be leveraged for therapeutic interventions in mental health care.

[19] ([19]) first published D&D in 1974, experiencing fluctuating popularity until its recent resurgence with the release of Stranger Things ([14]), actual-play podcasts like Critical Role ([28]), and films such as Dungeons & Dragons: Honor Among Thieves ([17]). The resurgence of D&D has cemented it as a cultural icon, influencing not only the gaming industry but also the entertainment and media sectors. D&D’s creation marked a radical departure from tabletop gaming ([19]). It introduced the concept of character role-play to board games, creating a new genre of games in the process: Tabletop Role-Playing Games (TTRPGs). Allowing players to embody their characters and collectively shape narratives through actions has been the cornerstone of D&D’s enduring appeal. Its legacy is a testament to the power of imagination and humanity’s desire to explore, interact, and tell stories within fantastical, abounding worlds ([10]).

Dungeons & Dragons (D&D) is a game of infinite possibilities, distinguished by its expansive rules and creative freedom, limited only by the imagination of players and the Dungeon Master (DM) ([18]). Played by groups of four to seven, excluding the DM, the game can take place either in person or online. The DM guides the adventure while players control their characters (PCs) and use self-created or pre-made character sheets to engage with the world. In-person sessions typically involve a gridded battle mat and miniatures to represent characters, with dice adding unpredictability to the experience ([42]). This flexible and immersive format has made D&D a powerful tool in therapy, offering players the opportunity to explore narratives, express emotions, and experiment with new behaviors in a safe, supportive environment ([1]; [4]; [7]; [8]; [12]; [20]; [18]; [21]; [24]; [25]; [34]).

City of Mist (CoM) ([29]) and Case: Amnesia Town ([30]) are other narrative-driven role-playing games that similarly use two six-sided dice and player-selected tags (such as powers and skills) to determine significant moves. The games feature a Master of Ceremonies (MC), and players choose how much of their character is mythological, selecting four themes split between mythos (representing fantastical elements, like a lawyer embodying Robin Hood) and logos (reflecting personal identity and routine). These themes define a character’s powers and personality within the story. During gameplay, players take actions when narratively appropriate, selecting relevant tags and rolling two six-sided dice. A roll of seven or higher results in varying levels of success, while a six or lower leads to failure, requiring an explanation of the consequences ([29], [30]).

Attachment theory explains how early caregiving relationships shape interpersonal patterns, which can be modified through awareness and therapy ([2]; [6]). Research shows that secure attachment fosters active engagement in experiential games, while insecure attachment can hinder collaboration ([16]; [38]). Dungeons & Dragons (D&D) provides a structured space for clients to explore attachment-related behaviors, practice emotional regulation, and develop healthier relationships. Through role-playing, players can experiment with trust, communication, and boundary-setting in a safe environment. By integrating RPGs into therapy, clinicians can help clients recognize and shift maladaptive attachment patterns. The game’s collaborative nature fosters social connection, emotional resilience, and personal growth, making it a valuable therapeutic tool.

Trauma overwhelms an individual’s internal resources, leaving lasting psychological symptoms stored in the limbic system as images, emotions, and sensations ([3]; [33]; [37]). When triggered, these memories resurface with the same intensity as the original event due to disrupted neural processing and limited emotional regulation. Creative interventions that engage non-verbal memory systems, like tabletop role-playing games (TTRPGs), offer alternative pathways for processing trauma. Dungeons & Dragons (D&D) facilitates trauma recovery by integrating storytelling, role-play, and social collaboration, allowing individuals to externalize trauma narratives and experiment with new identities in a safe, structured environment. By embodying characters, players can navigate emotional challenges, build resilience, and regain a sense of agency. Research supports the efficacy of TTRPGs in trauma therapy. Studies show that game-based cognitive behavioral therapy (GB-CBT) reduces trauma symptoms and improves emotional regulation in children ([34]), while psychodrama-based approaches empower survivors and enhance coping strategies ([32]). D&D’s interactive nature makes it a valuable therapeutic tool for trauma recovery.

Narrative therapy provides a theoretical framework emphasizing creativity by integrating language with creative and emotional engagement, accessing deep cognitive processes ([39]). TTRPGs utilize narrative transference, making them a compelling tool for therapeutic exploration. [41] ([41]) notes that games provide an opportunity for problem-solving and critical thinking by allowing participants to reflect on the narrative in their lives. [20] ([20]) further explore the concept of emergent narratives in TTRPGs, highlighting how game architecture, both physical and digital, facilitates player-driven storytelling. As structured yet flexible narrative-based games, Dungeons & Dragons (D&D) and TTRPGs offer a therapeutic space where clients can externalize personal struggles, experiment with new identities, and develop resilience through collaborative storytelling and character-driven problem-solving.

Similarly, Cognitive Behavioral Therapy incorporates creativity through role-playing and problem-solving to reframe negative thoughts ([9]), supporting cognitive restructuring and emotional regulation. Expressive arts therapy leverages creativity to help clients express emotions and experiences that are difficult to articulate verbally, using activities like art, music, drama, adventure, and play therapy ([13]). Humanistic theories, such as Carl Rogers’ Person-Centered Therapy, view creativity as essential for personal growth and self-discovery ([9]). Each of these theoretical frameworks provides a foundation for using creativity in therapy to process emotions and facilitate therapeutic change.

Dungeons & Dragons (D&D) balances player freedom with structured mechanics, allowing for dynamic decision-making within a controlled framework ([11]). The use of dice introduces unpredictability, mirroring real-life uncertainties, while character attributes—Strength, Dexterity, Constitution, Intelligence, Wisdom, and Charisma—serve as a framework for self-exploration and growth ([18]). Clients roll four dice six times, totaling up the highest three numbers from each roll, and then distribute these points across their character attributes. This process offers insight into their self-perception, as their choices may reflect underlying negative thoughts and beliefs about their strengths and weaknesses. These mechanics create an engaging space for therapeutic work, fostering resilience and adaptability.

Role-playing games (RPGs) have been shown to enhance self-awareness, social skills, and problem-solving abilities ([1]; [4]; [8]; [12]). The therapeutic relationship between the player and game master is key, providing a safe space for personal exploration and empowerment ([20]). Social withdrawal and impairment increase the risk of depression in adulthood ([23]); however, developing social skills through RPGs in a structured, low-risk environment may help mitigate this risk. D&D’s collaborative nature fosters belonging and shared problem-solving, reinforcing real-life interpersonal skills.

[15] ([15]) discusses the concept of *Reparative Play* in his article “Reparative Play in Dungeons & Dragons”, describing how players heal by asserting autonomy through their actions. He gives the example of how D&D can serve as a medium for queer individuals to create and affirm their identities. By setting the narrative of the game into a more accepting fantasy world, players engage in reparative play to give their characters a more accurate and positive representation of themself. This performativity helps players create their own progressive discourse, allowing them to imagine their world and experiences in a more positive light. These experiences in the game allow for cathartic experiences referred to in this paper as *emancipatory bleed*, which allow a player to feel a sense of liberation from societal constraints. Allowing the player the autonomy to navigate their character’s actions with intentionality can lead to a process referred to as *Liberatory Steering.* See the Transformative Play Model shown in Figure 1 as an example of how these interact. Games like D&D can be ideal in facilitating this creative experience. The latest edition of D&D further enhances this experience through two key elements: (1) affording players room to craft individual stories within a greater arching narrative and (2) encouraging the creative expression of personalized characters, oftentimes as an avatar of the player existing in a safe space ([11]).

While RPGs offer mental health benefits through escapism, they also pose risks if not carefully facilitated ([25]). Some studies highlight concerns about misinformation and problematic narratives in TTRPGs, but these can be mitigated through informed therapeutic approaches, as discussed in case studies in this article ([26]). Fantasy role-play provides a safe space for clients to explore complex issues, apply insights into real-life challenges, and reframe harmful stereotypes and biases, fostering a deeper understanding of relationships and social injustices. Additionally, RPGs promote creativity, self-expression, and personal growth ([27]). By addressing these concerns, therapists can ensure RPGs remain a valuable tool for healing and self-discovery.

Character creation in Dungeons & Dragons (D&D) fosters self-reflection, helping players recognize unconscious thought patterns and core beliefs. According to cognitive theory, automatic negative thoughts shape one’s self-perception, situation, and future ([5]). Assigning ability scores in D&D can mirror perceived strengths and weaknesses, offering therapists insight into clients’ self-concept and emotional struggles ([8]).

The game’s unpredictability, introduced through dice rolls, allows players to experience both setbacks and successes, fostering resilience ([18]). As clients engage with their characters, they externalize personal struggles and explore new coping strategies through role-play. This structured yet imaginative approach supports empowerment, self-awareness, and therapeutic growth ([10], [11]).

[21]’ ([21]) concept of “bleed” in Dungeons & Dragons (D&D) can be a powerful therapeutic tool, allowing players to process emotions through their characters. Bleed-in occurs when real-life emotions influence gameplay, providing a safe space for clients to explore anxiety, stress, or other challenges through their character’s actions. Conversely, bleed-out happens when in-game experiences shape real-life emotions, enabling players to internalize growth, resilience, and improved coping skills.

A skilled therapeutic game master can guide clients through these experiences, using techniques like aesthetic distancing to prevent negative bleed-out effects. Higher-level bleeds—Relationship, Emancipatory, and Identity—can further facilitate transformative shifts in self-perception and interpersonal dynamics, enhancing D&D’s potential as a tool for healing and personal development ([21]).

[22] ([22]) define shared pretensive realities as collaborative, adult equivalents of children’s pretend play, exemplified in the community-building nature of tabletop role-playing games (TTRPGs). [24]’s ([24]) meta-analysis highlights TTRPGs as immersive spaces for identity exploration, utopian discussions, and peer support, enhancing their therapeutic value for mental health.

TTRPGs also provide emotional support, as seen in [31] ([31]), where a gaming community created virtual spaces for a player’s terminally ill child, demonstrating their capacity for solidarity and comfort. Social bonds are further strengthened through shared participation, fostering trust, self-efficacy, and social skills, making TTRPGs valuable for treating social anxiety ([36]; [1]). As a narrative therapy tool, TTRPGs promote empathy, personal growth, and belonging, offering a dynamic approach to emotional exploration and healing ([7]).

Dungeons & Dragons (D&D) has played a pivotal role in shaping the tabletop role-playing game (TTRPG) genre, evolving from a niche hobby into a widely recognized cultural phenomenon. Its blend of storytelling, roleplay, and strategy has not only influenced media and entertainment but also demonstrated significant therapeutic potential. The game’s resurgence, fueled by popular media and actual-play content, has reinforced its value in fostering creativity, social engagement, and problem-solving. The structured yet flexible nature of D&D allows players to explore attachment patterns, process trauma, and develop resilience through narrative immersion and collaborative storytelling. Furthermore, TTRPGs as a whole provide a supportive space for identity exploration, emotional processing, and interpersonal skill development, making them valuable tools in therapeutic settings. The purpose of this article is to explore different ways to utilize TTRPGs in therapy, examining their potential for enhancing emotional regulation, social skills, and personal growth. By integrating role-playing mechanics into therapy, clinicians can leverage the transformative power of D&D to facilitate healing and meaningful psychological change.

## 2. Mastering Dragons: Case Examples Using RPGs in Therapy

### 2.1. Case Examples Using RPGs in Individual Therapy

All clients were informed of their rights in accordance with ethical standards in their respective states, ensuring informed consent and adherence to professional therapeutic guidelines. TTRPG rules were adapted based on session length, age, and therapeutic goals, with modifications to fit 50 min sessions for children and teens. Games were selected based on therapist and/or client familiarity and tailored to reflect the client’s presenting issues. Adjustments to character creation, such as modifying thematic elements, will be discussed further in the character creation section. As with game selection, the counselor determined the focus of each session based on the client’s current treatment goals.

#### 2.1.1. Kelsey (TTRPG: City of Mist)

Building upon the therapeutic applications of TTRPGs discussed earlier (e.g., [32]; [34]), this case study highlights how City of Mist ([29]) was used to address a young client’s trauma related to her parents’ divorce and subsequent remarriages, as well as her fear of seeing ghosts. Kelsey, a ten-year-old white female, had reported seeing both frightening and non-frightening ghosts, and her family’s cultural belief in such occurrences played a central role in shaping the counseling approach. In line with research on TTRPGs facilitating emotional engagement and self-reflection (e.g., [20]), the counselor introduced City of Mist as a narrative tool to help Kelsey process her fears in a structured and engaging way. The counselor adapted the game’s storyline, Case: Amnesia Town ([30]), to reflect the client’s experiences and fears, incorporating elements like ghosts while ensuring the narrative was age-appropriate.

The process began with the character creation session, which took approximately two 50 min sessions. Kelsey created two characters: one for her and one for her best friend. The client also had her mom join and watch her play the game. When the gameplay started, the client transferred her daily life to the story in a literal sense. For example, she and her best friend participated in sports and school as a main part of the plot.

[30] ([30]) and the City of Mist team created a story/case complete with characters, settings, and events related to nightmares coming to life entitled Case: Amnesia Town. The counselor had played this story with friends and was able to adapt the story to address the ghosts the client was seeing. The counselor acted as the MC and presented the case of someone using special effects to make the townspeople think they were seeing mean ghosts. The client chose to make her character a spy who received a mission to solve the mystery. Over the course of treatment, the client decoded the mystery, found the villain’s hideout, and defeated the villain with her “super-fast soccer ball”. The client high-fived her mom at the end of the story.

Over the course of four sessions, the game not only allowed Kelsey to engage in problem-solving and role-play through her character but also helped her work through the trauma in a safe, empowering context. By the conclusion of the story, Kelsey’s reported fear of ghosts had dissipated, with the counselor’s therapeutic use of TTRPGs contributing to her successful emotional regulation and symptom relief. The client’s mother confirmed Kelsey’s continued well-being over a year after therapy, further suggesting the long-term benefits of TTRPG interventions in trauma therapy.

#### 2.1.2. Samuel (TTRPG: City of Mist)

Building on the therapeutic potential of TTRPGs, as discussed in prior literature (e.g., [32]; [34]), Samuel’s case highlights the application of City of Mist ([29]) in processing relational trauma and emotional regulation. Samuel, a seven-year-old white male with adjustment disorder, struggled with anger, emotional dysregulation, and difficulty focusing, which were exacerbated by his father’s return to his life. Samuel reported to the counselor that he struggled with how to stay calm and was experiencing confusion about his father. The use of TTRPGs in this context provided a creative and engaging way to process complex feelings and relationships. After introducing Samuel to City of Mist, the counselor employed the game to help him explore and better understand his conflicted feelings toward his father.

(1)Character Creation: Samuel chose to base his character on the rift of a children’s show. Due to Samuel’s difficulties in staying focused, we chose to have one mythos (power) and one logos (routine) theme. Samuel’s logos was a school student with the tags: student, can skip class, Nino (friend), and makes friends easily. His mythos was a type of cat with the tags: protect everyone with cat-like abilities, claws out (transformation), and the ability to alter other’s allegiances. Samuel spent one counseling session creating his character, discussing the setting of the story, detailing other characters in the story, and drawing his character.(2)RPG Sessions: During the first session, the counselor invited the client to select a City of Mist (CoM; [29]) map as the starting setting, noting that while the map is optional, it serves as a helpful visual aid. The counselor focused on teaching the game’s rules, encouraging the client to roll for most actions while guiding the narrative forward. In this session, the client’s character confronted a villain in a museum and protected a friend. The counselor prompted the client to role-play various actions, such as fighting the villain, hiding from them, maintaining the character’s transformation, and escaping the museum unscathed.After the initial session, the client and counselor collaboratively developed the storyline, with the client selecting most of the key plot points. The central theme the client consistently revisited was the dual role of the character’s father as both a parent and a villain. In the narrative, the character knows his father is the villain, but the father is unaware that his archnemesis is his own son. The character’s mother had died in an accident caused by the father. In each session, the client either tried to confront his father or expose his true identity to others. The client often directed the counselor to play his character’s sidekick, guiding her actions throughout the story.The client advanced the storyline to when his character was an adult, now responsible for several adopted children and struggling with the challenges of raising them alone. The client revealed that the character’s father had passed away after they reconciled and learned each other’s identities. The character then expressed frustration, saying that “having a wife would make everything easier”. In the following session, the client returned to the original storyline, revisiting the moment before his character confronted his father. This time, the setting was slightly altered, with the client describing a bright room where his character’s mother’s body was kept. The father figure was portrayed as more sorrowful than angry, though he still sent other villains to attack the character.(3)Conclusions: Several takeaways emerged from the counselor’s observations of the client’s play. First, the client consistently returned to the father figure in the game, expressing confusion about how his character could be both a hero and a villain at the same time. Second, the client often incorporated recent life changes into the story, such as developing a crush at school, which led him to ask out characters in the game. Third, the client showed resistance to the game’s rules, providing opportunities for productive conversations about failure. For instance, when the client failed a roll, the counselor encouraged him to describe the outcome, prompting him to explain how his character failed (e.g., falling, missing the villain, being seen) and how the client resolved the issue to continue the story. In one instance, when the character failed a roll to sneak back home, the client described how his father caught him and how he explained himself without revealing his true identity.

In line with the narrative therapy framework in TTRPGs ([7]; [20]), Samuel’s character creation reflected his real-life struggles, focusing on themes of protection, transformation, and conflict. Over several sessions, the client not only processed his relationship with his father but also navigated frustration, failure, and emotional growth by engaging with the game mechanics, such as rolling for actions and developing the narrative. The evolving storyline mirrored Samuel’s emotional journey, offering a safe and controlled space to address complex family dynamics. The game served as a unique tool to foster coping skills, self-regulation, and problem-solving strategies, demonstrating the therapeutic potential of TTRPGs in addressing emotional and relational challenges in young clients.

#### 2.1.3. Ana (TTRPG: City of Mist)

Ana is a 12-year-old Latina female presenting with Dissociative Identity Disorder (DID) and Post-Traumatic Stress Disorder (PTSD). She had been in counseling for a year prior to using a TTRPG. Ana had a long history of abuse and had recently moved to a new home, which led to her seeking therapy. Over time, the counselor noticed specific changes in Ana from session to session, particularly her frequent confusion about what she had done at different times during the day. After extensive discussions, assessments, and consultations with Ana’s father (with the client’s permission), the counselor diagnosed her with DID. Ana was aware of her distinct “people” (as she referred to them) and began sharing details about her three identities. The counselor had the opportunity to interact with each of them. Ana expressed frustration with not knowing much about her people beyond the traumas they carried or kept hidden from her. To help her understand them better, the counselor proposed using a TTRPG, where each identity could have a character of their own. Ana agreed to the idea.

(1)Character Creation: Ana created three characters using the City of Mist (CoM) system ([29]). Since these characters represented distinct aspects of herself, the themes and tags are withheld to protect her anonymity. The character creation process involved selecting powers, skills, and weaknesses. For each identity, Ana developed a character with four themes, each containing three to four strengths and skills and one weakness. In total, Ana identified approximately 12–16 strengths and skills, along with 4 weaknesses for each of her people. This process allowed Ana to explore parts of herself she had previously felt anxious about, offering a fresh perspective. She expressed excitement about getting to know her people better and humorously remarked to the counselor that she might need a lanyard listing each person’s likes, strengths, and dislikes to share with others during interactions.(2)Conclusions: The client and counselor initially planned to begin a pre-written story from the City of Mist (CoM) system ([29]) and allow Ana to choose which character would respond or make moves in the game. However, due to another major family change, Ana had to terminate counseling before they could begin. Had the counselor continued working with Ana, the pre-written story would have been used, with space provided for Ana to consult with her different identities before deciding how her characters would react or act. This approach had already been practiced during sessions, where Ana would voice her need for time to sort out discussions between her people. The counselor facilitated this process by keeping a journal for Ana, stepping out of the room to allow time for the client to talk to her people or write down their thoughts. The counselor would then return to the room, where Ana would read her people’s responses to the counselor’s comments or situations. The counselor would either write back or engage in verbal dialogue, depending on what was requested.

Ana’s engagement with the TTRPG required a more careful setup to address her therapeutic goals of gaining insight into all aspects of herself. Establishing a trusting relationship with Ana and her identities was essential for facilitating open communication. One key rule was that all of Ana’s identities had to agree to participate in the game, ensuring that sensitive information would not be revealed by any of her identities without consent. This rule reinforced the importance of autonomy for each identity, supporting a collaborative, respectful approach to the therapeutic process. These considerations echo the findings from [7] ([7]) and [20] ([20]), who emphasize the need for careful attention to the dynamics of narrative therapy in TTRPGs, especially when working with complex psychological conditions like DID.

#### 2.1.4. Sally (TTRPG: Dungeons & Dragons)

Sally is a 19-year-old college student who enjoys art and music, particularly playing the piano. She spends time with her friends when not focused on schoolwork. Having moved away from home, she has experienced some challenges related to depression and anxiety. Sally reported that while her family is very supportive and she values their opinions, their reactions sometimes exacerbate her anxiety and depression. Before seeking counseling, Sally had not been in long-term therapy but had been dealing with medical concerns, notably high blood pressure. Her doctor recommended therapy to help manage stress and lower her blood pressure.

While Sally generally reported being content with her life, she was particularly worried about the changes in her habits following her medical diagnosis. She began avoiding leaving her house out of fear of contracting the COVID-19 virus. Previously, Sally had enjoyed game nights with friends and often participated in role-playing games like Dungeons & Dragons. However, due to the pandemic, her social opportunities were limited, and she became increasingly anxious about going out. In her initial counseling sessions, Sally shared that she was anxious about potential exposure to the virus.

Sally possessed several coping skills, including listening to music, drawing, and painting, as well as a desire to exercise. However, she struggled with low self-confidence, which hindered her ability to fully engage with these coping techniques.

At intake, Cognitive Behavioral Therapy (CBT) was selected as the most effective treatment for Sally’s symptoms, supplemented with art therapy to allow her to express inner conflicts she struggled to verbalize. While Sally showed mild improvement in the first four weeks, she remained preoccupied with worries about her medical uncertainty and blood pressure. During treatment, Sally shared that she continued to play Dungeons & Dragons (D&D) with friends once a week, prompting the addition of D&D sessions to her treatment plan. This adjustment was met with enthusiasm, and Sally appeared more energized and engaged. Over the next six weeks, D&D was fully incorporated into her sessions, alongside Sandtray and Narrative therapy, to enhance her therapeutic experience.

(1)Character Creation: During the initial sessions, Sally and the therapist discussed character creation, story, and scenery. Sally’s prior knowledge of D&D made this process smoother and more enjoyable, allowing the character creation process to closely mirror a real D&D campaign. For clients without prior D&D experience, this stage might require more explanation. Sally rolled for her character’s stats and was asked to rate how she perceived herself in these areas. She selected her character’s race, gender, and origin, and was then instructed to choose miniature figurines that represented her current feelings of anxiety and depression as accurately as possible. Although Sally selected several miniatures, most did not align with the challenges she described in her real life. Following character creation, the therapist and client worked together to arrange the figures on a sandtray, which was photographed for continuity. The journey then began.(2)RPG Sessions: During sessions three and four, Sally and the therapist advanced the story through a series of choices and reflective questioning. These middle sessions aimed to establish the narrative and encourage reflection and problem-solving. For instance, Sally was presented with the choice of facing a wandering monster, symbolizing an irrational thought specific to her, while on her way to a saving point in the story. She could choose to evade, avoid, or confront the monster, each option providing an opportunity for her to reflect on her decision. Sally opted to engage the monster and play out the encounter using her coping skills as her character moved forward. Dice rolls were used to introduce unpredictability; if Sally rolled low, she had to quickly adapt by thinking of alternative actions or pausing to ground herself and regain control. These encounters continued throughout both sessions, offering a reflective look at Sally’s ability to apply her coping skills while exploring the depth of her anxiety. The guidance of the narrator in her own story, while maintaining the magical and enjoyable elements, proved beneficial. By the end of these sessions, Sally reported a decrease in anxiety and felt she was gaining a deeper understanding of it.These sessions proved to be some of the most therapeutic for Sally throughout her counseling experience. As she progressed through her story, she reached the climactic boss fight—yet unlike traditional D&D scenarios with formidable villains, she found herself in a loot room surrounded by treasures. However, what she did not anticipate was the presence of a mirror reflecting her own image. In this phase, the therapist guided Sally to confront her self-perceived faults. Through carefully crafted questioning, elements of Cognitive Behavioral Therapy (CBT), Choice Theory, and Narrative Therapy were seamlessly integrated into the game world.Sally faced her biggest enemy and greatest strength: her self-esteem. As she questioned her reflection, she began noticing subtle clues—observations the therapist had previously made and incorporated into the game to provide insight. Techniques from Solution-Focused Brief Therapy (SFBT), such as coping questions and presupposing change questions, helped shift her perspective. From CBT, the therapist introduced in-game monsters representing cognitive distortions, along with exposure elements and cognitive restructuring, to foster healthier problem-solving. As Sally recognized these patterns, her awareness deepened, both within the game and in her real life.This mental breakthrough, which Sally had not anticipated, caught her completely off guard. At first, she had been simply playing a game, unaware of how closely the story mirrored her real-life challenges and obstacles. These final sessions emphasized Sally’s ability to connect the dots and integrate the insights she had gained. The conclusion of her campaign came when she used an in-game magical item to trap her reflection, carrying it with her wherever she went. After the sixth session, Sally rated her anxiety lower than in previous weeks. She also reported a decrease in her depression and medical worries, while her self-esteem rating improved. Reflecting on her triumph, Sally shared that she had never fully recognized how “real and controlling” her anxiety had been in her life. She came to see how her anxiety had influenced her decisions, either pushing her to act or preventing action altogether. She realized that eliminating her anxiety, rather than learning to coexist with it, would only lead her back to the “loot room”.(3)Conclusions: Although the client demonstrated some initial improvement with the original treatment plan, which included CBT and art therapy, it was not until the incorporation of D&D role-playing that she became fully invested in her treatment. This addition facilitated her immersion in learning coping skills, developing new beliefs, and practicing problem-solving strategies in a meaningful and engaging way. The established rapport between client and therapist played a crucial role in this process, providing a foundation of trust that allowed the narrative approach within the D&D campaign to flourish. This collaborative dynamic made the therapeutic experience feel more organic and individualized, extending beyond a standard manual-based intervention ([22]; [24]). By combining narrative elements with in-game decision-making, the therapist was able to guide the client through complex emotional challenges in a way that felt both safe and transformative, enhancing the therapeutic impact.

### 2.2. Case Examples Using RPG in Group Therapy

#### 2.2.1. RPGs with Autism Spectrum Disorder

In therapeutic settings, particularly when working with individuals with Autism Spectrum Disorder (ASD), adapting materials to meet diverse needs and preferences is essential for fostering engagement and success. This was exemplified in the case of “Egg”, a participant in our Dungeons & Dragons (D&D) therapy group, who played a human wizard. Recognizing Egg’s desire to reduce cell phone usage and his preference for tangible materials, a tailored approach was necessary to ensure his engagement with the game.

D&D, especially for first-time players, involves managing complex character details, including class abilities, magical spells, and game mechanics. While digital platforms effectively organize this information, they conflicted with Egg’s goal of minimizing screen time. To address this, we created simplified character sheets that transferred essential information to paper and utilized a website to print spell cards for easy reference during gameplay. Furthermore, recognizing Egg’s tactile sensitivity to paper, all materials were encased in plastic protector sheets, allowing him to track changes with a dry-erase marker, a feature he found particularly helpful.

Beyond individual accommodations, the therapy group aimed to enhance social skills through structured D&D interactions. This unique setting provided opportunities for emotional and social development. Participants, including “Blood Owl” and Egg, embodied aspects of their real-life personalities—Blood Owl, an impulsive assassin, and Egg, an analytical wizard. Blood Owl’s in-game behavior, reflecting his challenges with impulsivity and excitement regulation, presented therapeutic opportunities for growth.

A pivotal moment occurred when Blood Owl’s actions frustrated Egg, prompting Egg to express his feelings about how the behavior affected the group. This vulnerable interaction led to Blood Owl apologizing and adjusting his actions, demonstrating a newfound awareness of the group’s dynamics. Assessments using the Warwick–Edinburgh Mental Well-Being Scale (WEMWBS; [35]) indicated improvement in Blood Owl’s mental well-being, further reinforced by positive family feedback.

The case of Blood Owl and Egg illustrates the potential of D&D as a therapeutic tool for enhancing social skills, empathy, and emotional learning in adolescents with ASD. These findings highlight the importance of flexibility and creativity in therapeutic settings. By tailoring materials to meet individual needs, therapists can enhance accessibility and foster a more inclusive therapeutic environment. This approach aligns with the growing body of literature suggesting that personalized adaptations in therapeutic interventions, such as role-playing games, can improve outcomes and increase inclusivity ([7]; [4]; [26]; [31]).

#### 2.2.2. RPGs with LGBTQ Adolescents

Platforms like D&D Beyond ([40]) can streamline the character creation process in Dungeons & Dragons (D&D), offering clinicians not only a practical tool for campaign management but also a gateway for clients to explore and express aspects of their identity in a symbolic and empowering way. This is particularly valuable in therapeutic groups focused on identity development, such as those for LGBTQ+ adolescents. In these settings, the choices surrounding a character’s race, class, and background are not merely whimsical; they often serve as profound reflections of personal experiences, struggles, and aspirations.

For example, the creation of a Tiefling warlock named “Lilith” by a transgender female group member highlights how character creation can be an allegorical expression of identity. In D&D, Tieflings are often depicted as outcasts, bearing physical traits that evoke fear and prejudice—traits that mirror the societal stigma faced by transgender individuals. Just as Tieflings are judged based on their appearance and heritage, transgender individuals frequently encounter discrimination for aspects of their identity that are inherent and unchosen. The choice of a Tiefling character by a transgender participant serves as a form of narrative therapy, where the fantasy setting provides a safe space to explore and confront issues of identity, acceptance, and resilience.

In this context, the Tiefling character’s journey is a powerful metaphor for the transgender experience, as it emphasizes that one’s background or birth circumstances do not define their potential or moral compass. The Tiefling’s ability to act beyond societal judgment offers a transformative opportunity to explore how marginalized individuals might respond to prejudice. This narrative exploration fosters empowerment, helping clients navigate challenges in a controlled, supportive environment where they can experiment with responses to discrimination and prejudice.

This therapeutic tool aligns with the article’s purpose of showcasing how role-playing games, like D&D, provide a platform for identity exploration and emotional learning. By allowing clients to create characters that resonate with their personal stories, clinicians can facilitate deeper engagement and self-reflection. These allegorical connections enable clients to process emotions, develop coping strategies, and build a stronger sense of self. This aligns with the growing body of literature suggesting that therapeutic role-playing games can be effective tools in promoting identity development, particularly in marginalized groups ([24]).

#### 2.2.3. RPGs with Residential Drug Treatment

This section explores the use of Dungeons & Dragons (D&D) as a therapeutic tool in a residential drug treatment facility for women. The facility’s clients, who presented with a range of diagnoses—including depression, anxiety, bipolar I disorder, and borderline personality disorder—shared a common experience of being in recovery for Substance Use Disorder (SUD) and often had histories of sexual and/or domestic assault. The substance dependencies varied, encompassing alcohol, methamphetamine, heroin, and fentanyl.

A particularly notable dynamic within the group was the ongoing conflict between two women, which had escalated to a level that required intervention from the facility’s executive director. The introduction of D&D as a therapeutic medium significantly shifted the group dynamics. After the conflict, both women agreed to participate in a D&D session, where the therapeutic game master intentionally designed encounters that emphasized cooperative problem-solving and teamwork, aiming to facilitate collaboration between the two women. The specific puzzle the clinician used in-game is referred to as a “balance scale”. A room containing a giant golden scale required the two characters to help each other onto the weighted platforms and trade items back and forth from their inventory until the scale balanced, unlocking a secret room containing a treasure chest. To further collaboration, the clinician made the women roll sleight-of-hand checks to ensure that their characters caught the thrown items. This scenario was meant to be mildly distressing to the players, making their collaboration necessary to solve the puzzle and be rewarded. Remarkably, within just half an hour of gameplay, the atmosphere transformed from tension to cooperation, laughter, and mutual enjoyment. This shift underscores the unique capacity of D&D to foster teamwork and interpersonal growth, as the game’s inherent structure requires players to work together towards a shared goal.

Over the course of two and a half months, the group participated in eight D&D sessions alongside their individual therapy and psychoeducational group sessions. One notable observation was that the clients expressed a distinct preference for the combat aspects of the game over role-playing. They found empowerment and satisfaction in overcoming in-game challenges, vanquishing enemies, and achieving shared objectives. This preference can be interpreted as a reflection of their desire to regain a sense of control and agency—both of which are often compromised by the trauma of substance use and assault. The structured, safe environment of D&D allowed the women to explore themes of power and agency while contributing positively to their recovery process.

This case exemplifies the versatility of D&D as a therapeutic tool. It demonstrates how carefully structured group activities, like collaborative fantasy games, can not only help mitigate interpersonal conflicts but also contribute to empowerment and the broader recovery process for individuals struggling with SUD and trauma. The therapeutic use of D&D aligns with the growing body of literature that highlights the game’s potential to foster emotional development, build social skills, and empower individuals in treatment for mental health and substance use ([7]; [26]).

TTRPGs have been utilized in various therapeutic contexts, offering a versatile framework for psychological growth and emotional processing. Case studies have demonstrated their effectiveness in helping individuals develop problem-solving skills, process trauma, experience empowerment, navigate challenges such as failure and rule-breaking, and explore complex relationship dynamics. Additionally, TTRPGs support coping skill development, self-regulation, and identity exploration. In group settings, special needs, adolescent challenges, and substance use disorders have benefited from the structured yet flexible nature of TTRPGs. These games provide a supportive environment accommodating individual sensitivities while fostering social skills, empathy, emotional learning, identity exploration, and conflict resolution abilities. As a dynamic and immersive intervention, TTRPGs serve as a powerful tool for fostering growth, resilience, and healing.

## 3. Discussion

The use of tabletop role-playing games (TTRPGs) in therapy is gaining recognition for their ability to support clients facing a range of mental health challenges. Games like Dungeons & Dragons (D&D) create an immersive space where participants can develop coping skills, explore different aspects of their identity, and practice emotional regulation through character-driven narratives. TTRPGs have been utilized in diverse therapeutic contexts, demonstrating their effectiveness in fostering psychological growth and resilience. Case studies highlight their role in enhancing problem-solving abilities ([1]), processing trauma ([34]), and fostering empowerment through narrative agency ([20]). Additionally, these games provide a structured yet flexible framework for addressing complex life challenges such as failure, breaking rules, and navigating interpersonal relationships ([12]; [7]).

In group settings, TTRPGs have shown particular benefits for clients with Autism Spectrum Disorder (ASD), special needs, adolescent concerns, and substance use disorders. The structured yet adaptable nature of these games allows participants to engage in social interactions within their comfort zones, develop empathy, enhance emotional regulation, and strengthen identity exploration ([8]; [25]). Research also suggests that TTRPGs support the development of conflict resolution skills, self-regulation, and coping mechanisms in both individual and group therapy contexts ([4]; [38]). As an adaptable and immersive intervention, TTRPGs serve as a powerful therapeutic tool for fostering emotional growth, social learning, and psychological healing.

While these five case studies provide valuable insights into the potential effects of the treatment, we acknowledge the inherent limitations of this case study methodology. Given the absence of controlled conditions, alternative explanations for patient improvement cannot be ruled out, and findings should be interpreted as preliminary indications rather than definitive conclusions. Additionally, the sample size is small, limiting the generalizability of results. Another important consideration is potential biases related to both therapists and clients who were already familiar with TTRPGs and these particular formats, as prior exposure and enthusiasm for role-playing games may have influenced engagement and perceived benefits. Furthermore, participants’ predisposition for creative outlets may mean that TTRPG-based therapy is particularly effective for certain populations but not universally applicable. Addressing these limitations will require more rigorous methodologies, including randomized controlled trials and larger, more diverse participant groups.

Future research should explore the broader applicability of TTRPGs across diverse populations and therapeutic settings. Longitudinal studies could examine the lasting effects of role-playing on identity formation and mental health, while comparative research could clarify the unique contributions of TTRPGs relative to other therapeutic modalities, such as cognitive-behavioral therapy (CBT). Additionally, investigating the effectiveness of online role-playing games as a therapeutic tool is crucial, particularly as technology continues to transform social interaction and mental health care. Research from different cultural spheres and societies would also be valuable, as the rapid diversification of role-playing game formats presents new opportunities to assess their therapeutic potential across various contexts. Further inquiry should explore why and how TTRPGs might be preferable to other forms of socializing or structured interventions like CBT, particularly for individuals who may find traditional therapy intimidating or unappealing. Beyond autism, this approach may be especially beneficial for individuals who enjoy games, those skeptical of conventional therapy, and those at risk of early attrition in treatment due to a lack of intrinsic motivation. By identifying the populations for whom TTRPGs may be most effective, future studies can help refine and optimize their use as a therapeutic tool.

## Figures and Tables

**Figure 1 behavsci-15-00441-f001:**
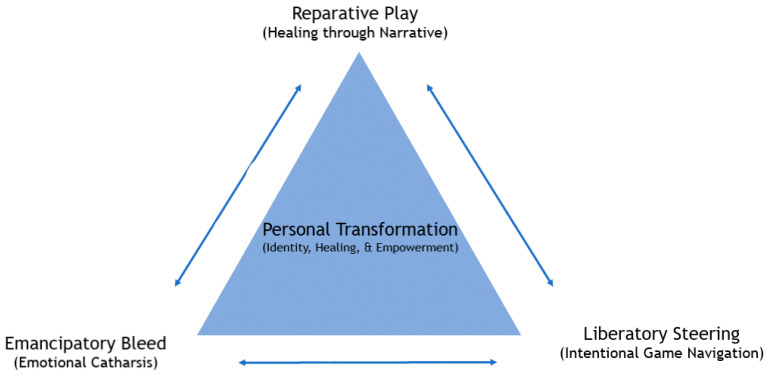
Transformative Play Model.

## Data Availability

No new data were created or analyzed in this study. Data sharing does not apply to this article.

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
