# Peer review of "Mastering Your Dragons: Using Tabletop Role-Playing Games in Therapy"

_behavsci, 2025, doi:10.3390/bs15040441_

Round 1
Reviewer 1 Report
Comments and Suggestions for Authors
This work is interesting and original even if, for example, a John Hopkins University web page of December 2023 reports Tabletop Therapy: How Dungeons & Dragons can improve mental health (approach to group counseling that combines role-playing with traditional therapy introduced in the spring of 2020). The authors in this paper report 5 case studies, but it is known that with this method, you can only obtain some indications and inaccurate conclusions on the effect of the treatment since there could be alternative explanations for the improvement of the patients. It would be helpful to experimentally treat this technique to be able to conclude that it is effective. In conclusion, the paper needs minor revisions since it would be appropriate to clarify the limits of the method.
Author Response
Thank you for your feedback on our article. We believe the suggested changes have strengthened the piece. Our responses and changes are below.
- Comment 1: The authors in this paper report 5 case studies, but it is known that with this method, you can only obtain some indications and inaccurate conclusions on the effect of the treatment since there could be alternative explanations for the improvement of the patients.
- Response 1: We definitely agree. As such, we have added a paragraph further clarifying the limitations of this study. This is found on page 13, lines 642-654.
- “While these five case studies provide valuable insights into the potential effects of the treatment, we acknowledge the inherent limitations of case study methodology. Given the absence of controlled conditions, alternative explanations for patient improvement cannot be ruled out, and findings should be interpreted as preliminary indications rather than definitive conclusions. Additionally, the sample size is small, limiting the generalizability of results. Another important consideration is potential biases related to both therapists and clients who were already familiar with TTRPGs and these particular formats, as prior exposure and enthusiasm for role-playing games may have influenced engagement and perceived benefits. Furthermore, participants’ predisposition for creative outlets may mean that TTRPG-based therapy is particularly effective for certain populations but not universally applicable. Addressing these limitations will require more rigorous methodologies, including randomized controlled trials and larger, more diverse participant groups.”
- Comment 2: It would be helpful to experimentally treat this technique to be able to conclude that it is effective.
- Response 2: You are right that more research is needed to ascertain whether TTRPGs are effective. To that ends, we revised and expanded the future research section. This is on page 14, lines 655-670.
- “Future research should explore the broader applicability of TTRPGs across diverse populations and therapeutic settings. Longitudinal studies could examine the lasting effects of role-playing on identity formation and mental health, while comparative research could clarify the unique contributions of TTRPGs relative to other therapeutic modalities, such as cognitive-behavioral therapy (CBT). Additionally, investigating the effectiveness of online role-playing games as a therapeutic tool is crucial, particularly as technology continues to transform social interaction and mental health care. Research from different cultural spheres and societies would also be valuable, as the rapid diversification of role-playing game formats presents new opportunities to assess their therapeutic potential across various contexts. Further inquiry should explore why and how TTRPGs might be preferable to other forms of socializing or structured interventions like CBT, particularly for individuals who may find traditional therapy intimidating or unappealing. Beyond autism, this approach may be especially beneficial for individuals who enjoy games, those skeptical of conventional therapy, and those at risk of early attrition in treatment due to a lack of intrinsic motivation. By identifying the populations for whom TTRPGs may be most effective, future studies can help refine and optimize their use as a therapeutic tool.”
- Comment 3: In conclusion, the paper needs minor revisions since it would be appropriate to clarify the limits of the method.
- Response 3: Again, we absolutely agree and have added the paragraph in Response 1 to expand on the limitations of this study.
Reviewer 2 Report
Comments and Suggestions for Authors
This article demonstrates with compelling evidence the benefits of role playing games as therapeutic tools especially when used together with other therapeutic tools. The cases presented in the article are well chosen and can help to think about what kind of combination of therapeutic tools would be most promising especially with target groups that are identified in the article. Personally I was a bit surprised that there was so little discussion about the potential (as well as differences with table top gaming) that online role playing games would provide - but that might be something for future research to analyze
Comments:
The most impressing strengths of this article are related to case studies where the benefits of combining role playing games with other therapeutic tools and as a means to facilitate friendly social transactions with other people. There obviously is great further potential for different kind of role playing games as therapeutic tools. This article introduced some of the pioneering games and their evolution. Since role playing games have in many societies become very popular and at the same time have diversified a lot, it would be interesting see more research where the therapeutic uses and potential are being critically examined. Moreover, it would be interesting see more research from different cultural spheres/ societies where these issues are studied. For instance, role playing games in the United States and Japan would have a quite different social context. Anyway, the role playing games as therapeutic tools and in combination with other tools should always be chosen carefully to suit the needs of the individual(s).
The online environment provides huge opportunities for modifying every aspect of any particular role playing games to suit the needs of individuals. Moreover, online environment also opens possibilities as well as risks when it comes to social interactions. That increases the role of the controller but also adds new opportunities. In other words, the present article serves well as an introduction to the earlier stages of using role playing games as therapeutic tools but since role playing games have rapidly diversifying due to technological development it would be natural to thing the opportunities provided by the huge variety of present and future role plying game alternatives.
This article serves its purpose well in the way it has been written and there is no particular reason to ask for major or minor revisions. The main strength of this article lies in the well presented case studies and those are firmly connected with the chosen method of presenting the benefits of the pioneering type of role playing games as therapeutic tools. However, my comments are simply provided to give ideas how the themes and ideas presented in the present article could be used in future research. It could be possible to add some discussion on these future prospects in the latter parts of the article. This article is fine the way it is written but the authors could easily continue to do research on other types of role playing games and their present or potential uses as therapeutic tools.
Author Response
Thank you for your feedback on our article. We believe the suggested changes have strengthened the piece. Our responses and changes are below.
- Comment 1: Personally I was a bit surprised that there was so little discussion about the potential (as well as differences with table top gaming) that online role playing games would provide - but that might be something for future research to analyze.
- Response 1: We agree that online role playing games have increased immensely and this is a valuable platform for future study. We revised and added this to the future research section. This can be found on page 14, lines 659-664.
- “Additionally, investigating the effectiveness of online role-playing games as a therapeutic tool is crucial, particularly as technology continues to transform social interaction and mental health care. Research from different cultural spheres and societies would also be valuable, as the rapid diversification of role-playing game formats presents new opportunities to assess their therapeutic potential across various contexts.”
- Comment 2: Since role playing games have in many societies become very popular and at the same time have diversified a lot, it would be interesting see more research where the therapeutic uses and potential are being critically examined. Moreover, it would be interesting see more research from different cultural spheres/ societies where these issues are studied.
- Response 2: We were not aware of how TTRPGs are used in other cultures or societies. That is a limitation and we added a discussion about future research opportunities with other cultures. This is on page 14, lines 655-670.
- “Future research should explore the broader applicability of TTRPGs across diverse populations and therapeutic settings. Longitudinal studies could examine the lasting effects of role-playing on identity formation and mental health, while comparative research could clarify the unique contributions of TTRPGs relative to other therapeutic modalities, such as cognitive-behavioral therapy (CBT). Additionally, investigating the effectiveness of online role-playing games as a therapeutic tool is crucial, particularly as technology continues to transform social interaction and mental health care. Research from different cultural spheres and societies would also be valuable, as the rapid diversification of role-playing game formats presents new opportunities to assess their therapeutic potential across various contexts. Further inquiry should explore why and how TTRPGs might be preferable to other forms of socializing or structured interventions like CBT, particularly for individuals who may find traditional therapy intimidating or unappealing. Beyond autism, this approach may be especially beneficial for individuals who enjoy games, those skeptical of conventional therapy, and those at risk of early attrition in treatment due to a lack of intrinsic motivation. By identifying the populations for whom TTRPGs may be most effective, future studies can help refine and optimize their use as a therapeutic tool.”
- Comment 3: Moreover, online environment also opens possibilities as well as risks when it comes to social interactions. That increases the role of the controller but also adds new opportunities. In other words, the present article serves well as an introduction to the earlier stages of using role playing games as therapeutic tools but since role playing games have rapidly diversifying due to technological development it would be natural to thing the opportunities provided by the huge variety of present and future role plying game alternatives.
- Response 3: Absolutely. We address this in the revised future research section added above in Response 2.
Reviewer 3 Report
Comments and Suggestions for Authors
The study describes a novel and engaging approach to game-based cognitive behavioural therapy and, as such, is well suited to the journal. Moreover, the case study methodology matches the exploratory nature of the topic, and the diversity in narratives does an excellent job showcasing the different people for whom roleplaying might have a therapeutic benefit.
In my view, the aim of the paper is laudable. However, given the niche area of study, it might be a difficult sell for clinicians who are unfamiliar with or sceptical of game-based therapy for adolescents and adults. To help mitigate this concern, I’d like to suggest a few additions to the Introduction:
- The focus is exclusively on D&D, whereas half the case studies utilise City of Mist. While it makes sense to prioritise D&D given its long history and recent cultural resurgence, it may be more instructive for the reader to learn about roleplaying in general, followed by a brief discussion of the various games chosen for the study and how they differ—if, indeed, there are any differences likely to influence the therapeutic effect. I acknowledge City of Mist is briefly introduced later in section 2.1, but I suggest moving this to the introduction so as to not surprise the reader, and to contrast it with D&D.
- I would be interested to know the rationale behind why D&D and City of Mist were chosen for the study, rather than say beginner friendly RPGs (e.g., Quest) or those with a stronger narrative focus (e.g., Fate). Was it simply participant interest or something more? The former is understandable, but it be nice to know what the authors’ intention was.
- Potential therapeutic mechanisms are discussed, which is quite useful, but I would recommend adding a framework for how RPGs should be applied in therapy. Was there a particular approach to implementation in the case studies, or were these RPGs run purely as entertainment and the therapeutic effects arose incidentally? For instance, it’s alluded to on line 417 that the therapist’s observations were important for priming realisations that emerged later during gameplay, and on line 528 that encounters were designed with the express purpose of facilitating problem-solving and teamwork, but it would be helpful to know if this was serendipitous of if there was a more systematic approach at play.
- One objection that may be encountered is that roleplaying games with fantastical settings might give rise to problematic behaviour if such behaviour is carried over to the real world. In fact, it’s mentioned on line 123 that “problematic narratives within TTRPGs … can be mitigated through informed therapeutic approaches”, but no information is given as to how this can be accomplished. Addressing this point in detail would help allay these concerns (if you’re interested, a good discussion for the beneficial use of fantasy can be found here: https://doi.org/10.1007/978-3-319-95681-7_80).
- Lastly, there were a few lines I felt could use a bit more detail:
a. A citation would be helpful to support the claim on line 69 that the “flexible and immersive format has made D&D a powerful tool in therapy…”.
b. On line 111, it wasn’t clear to me how “…character attributes—Strength, Dexterity, Constitution, Intelligence, Wisdom, and Charisma—offer a framework for self-exploration and growth”, so I would suggest adding some clarification for this point.
c. I appreciated that you mentioned social skills on Line 115, as that is a clear advantage of the structured interactions RPGs can provide. To build on this, you might mention that social withdrawal is believed to create a feedback cycle that worsens depression (https://doi.org/10.1007/s10802-011-9537-z), so RPGs can break this cycle by acting as a method for scaffolding social skills in a regulated, consequence-free space.
Another addition I’d like to see is in the Discussion. On line 585, it’s mentioned that a limitation of the present study is that comparative value against other interventions hasn’t been tested. To guide future research, I would encourage you to make suggestions as to why and how RPGs might be preferable to other forms of socialising or CBT. Aside from autism, are there other people for whom this approach could work better than traditional therapy? For instance, I imagine it could be more appealing to people who like games, less intimidating to people who are sceptical or uncomfortable in traditional therapy, and may help with attrition as there is an added incentive for continuation.
Lastly, there is a small typo in the DOI for reference 30.
The manuscript is excellent overall, however, and touches upon a truly exciting research area. The writing style is clear, and the results provide rich detail for application. With a few minor revisions, I believe this paper would make a superb publication that will enrich therapeutic practice.
Author Response
Thank you for your feedback on our article. We believe the suggested changes have strengthened the piece. Our responses and changes are below.
- Comment 1: The focus is exclusively on D&D, whereas half the case studies utilise City of Mist. While it makes sense to prioritise D&D given its long history and recent cultural resurgence, it may be more instructive for the reader to learn about roleplaying in general, followed by a brief discussion of the various games chosen for the study and how they differ—if, indeed, there are any differences likely to influence the therapeutic effect. I acknowledge City of Mist is briefly introduced later in section 2.1, but I suggest moving this to the introduction so as to not surprise the reader, and to contrast it with D&D.
- Response 1: That is a great suggestion. It enhances the article’s flow. We integrated it into the introduction and adjusted it accordingly. This change can be found on page 2, lines 75-85.
- “City of Mist (CoM) (Moshe, 2017) and Case: Amnesia Town (Moshe, 2020) are other narrative-driven role-playing games that similarly use two six-sided dice and player-selected tags (such as powers and skills) to determine significant moves. The games feature a Master of Ceremonies (MC), and players choose how much of their character is mythological, selecting four themes split between mythos (representing fantastical elements, like a lawyer embodying Robin Hood) and logos (reflecting personal identity and routine). These themes define a character’s powers and personality within the story. During gameplay, players take actions when narratively appropriate, selecting relevant tags and rolling two six-sided dice. A roll of seven or higher results in varying levels of success, while a six or lower leads to failure, requiring an explanation of the consequences (Moshe, 2017; Moshe, 2020).”
- Comment 2: I would be interested to know the rationale behind why D&D and City of Mist were chosen for the study, rather than say beginner friendly RPGs (e.g., Quest) or those with a stronger narrative focus (e.g., Fate). Was it simply participant interest or something more? The former is understandable, but it be nice to know what the authors’ intention was.
- Response 2: This is an important discussion. We added some information prior to case examples in individual therapy. We also added information in the limitations section about this.
- “All clients were informed of their rights in accordance with ethical standards in their respective states, ensuring informed consent and adherence to professional therapeutic guidelines. TTRPG rules were adapted based on session length, age, and therapeutic goals, with modifications to fit 50-minute sessions for children and teens. Games were selected based on therapist and/or client familiarity and tailored to reflect the client’s presenting issues. Adjustments to character creation, such as modifying thematic elements, will be discussed further in the character creation section. As with game selection, the counselor determined the focus of each session based on the client's current treatment goals.” [This is on page 5, lines 232-240.]
- “Another important consideration is potential biases related to both therapists and clients who were already familiar with TTRPGs and these particular formats, as prior exposure and enthusiasm for role-playing games may have influenced engagement and perceived benefits. Furthermore, participants’ predisposition for creative outlets may mean that TTRPG-based therapy is particularly effective for certain populations but not universally applicable. Addressing these limitations will require more rigorous methodologies, including randomized controlled trials and larger, more diverse participant groups.” [This is on page 13, lines 647-654.]
- Comment 3: Potential therapeutic mechanisms are discussed, which is quite useful, but I would recommend adding a framework for how RPGs should be applied in therapy. Was there a particular approach to implementation in the case studies, or were these RPGs run purely as entertainment and the therapeutic effects arose incidentally? For instance, it’s alluded to on line 417 that the therapist’s observations were important for priming realisations that emerged later during gameplay, and on line 528 that encounters were designed with the express purpose of facilitating problem-solving and teamwork, but it would be helpful to know if this was serendipitous of if there was a more systematic approach at play.
- Response 3: Thank you for pointing this out. It is very important for the goal of this article to provide the therapeutic justification and basis for decisions in order to help other clinicians understand and utilize these treatments. We have added the following to help explain.
- “As with game selection, the counselor determined the focus of each session based on the client's current treatment goals.” [This can be found on page 5, lines 238-240.]
- “Techniques from Solution-Focused Brief Therapy (SFBT), such as coping questions and presupposing change questions, helped shift her perspective. From CBT, the therapist introduced in-game monsters representing cognitive distortions, along with exposure elements and cognitive restructuring, to foster healthier problem-solving.” [This is on page 10, lines 465-469.]
- “The specific puzzle the clinician used in-game is referred to as a “balance scale.” A room containing a giant golden scale required the two characters to help each other onto the weighted platforms and trade items back and forth from their inventory until the scale balanced, unlocking a secret room containing a treasure chest. To further collaboration, the clinician made the women roll sleight-of-hand checks to ensure that their characters caught the thrown items. This scenario was meant to be mildly distressing to the players, making their collaboration necessary to solve the puzzle and be rewarded.” [This can be found on page 12, lines 581-588.]
- Comment 4: One objection that may be encountered is that roleplaying games with fantastical settings might give rise to problematic behaviour if such behaviour is carried over to the real world. In fact, it’s mentioned on line 123 that “problematic narratives within TTRPGs … can be mitigated through informed therapeutic approaches”, but no information is given as to how this can be accomplished. Addressing this point in detail would help allay these concerns (if you’re interested, a good discussion for the beneficial use of fantasy can be found here: https://doi.org/10.1007/978-3-319-95681-7_80).
- Response 4: Thank you for this resource. We revised this paragraph adding pertinent information. This can be found on page 4, lines 174-180.
- “While RPGs offer mental health benefits through escapism, they also pose risks if not carefully facilitated (Larche et al., 2021). Some studies highlight concerns about misinformation and problematic narratives in TTRPGs, but these can be mitigated through informed therapeutic approaches, as discussed in case studies in this article (Lis et al., 2015). Fantasy role-play provides a safe space for clients to explore complex issues, apply insights to real-life challenges, and reframe harmful stereotypes and biases, fostering a deeper understanding of relationships and social injustice. Additionally, RPGs promote creativity, self-expression, and personal growth (Mackenzie et al., 2020). By addressing these concerns, therapists can ensure D&D remains a valuable tool for healing and self-discovery.”
- Comment 5: A citation would be helpful to support the claim on line 69 that the “flexible and immersive format has made D&D a powerful tool in therapy…”.
- Response 5: Good point. We added citations to support this statement. This can now be found on page 2, lines 71-73.
- “This flexible and immersive format has made D&D a powerful tool in therapy, offering players the opportunity to explore narratives, express emotions, and experiment with new behaviors in a safe, supportive environment (Abbott et al., 2021; Baker et al., 2022; Causo & Quinlan, 2021; Clarke et al., 2019; Daniau, 2016; Henrich & Worthington, 2021; Gutierrez, 2017; Hugaas, 2022; Kawitzky, 2020; Larche et al., 2021; Springer et al., 2012).”
- Comment 6: On line 111, it wasn’t clear to me how “…character attributes—Strength, Dexterity, Constitution, Intelligence, Wisdom, and Charisma—offer a framework for self-exploration and growth”, so I would suggest adding some clarification for this point.
- Response 6: We added some more context to this paragraph to hopefully add clarification to this. This can be found on page 3, lines 132-141.
- “Dungeons & Dragons (D&D) balances player freedom with structured mechanics, allowing for dynamic decision-making within a controlled framework (Crawford et al., 2014b). The use of dice introduces unpredictability, mirroring real-life uncertainties, while character attributes—Strength, Dexterity, Constitution, Intelligence, Wisdom, and Charisma—serve as a framework for self-exploration and growth (Gutierrez, 2017). Clients roll four dice six times, total the highest three from each roll, and then distribute these points across their character attributes. This process offers insight into their self-perception, as their choices may reflect underlying negative thoughts and beliefs about their strengths and weaknesses. These mechanics create an engaging space for therapeutic work, fostering resilience and adaptability.”
- Comment 7: I appreciated that you mentioned social skills on Line 115, as that is a clear advantage of the structured interactions RPGs can provide. To build on this, you might mention that social withdrawal is believed to create a feedback cycle that worsens depression (https://doi.org/10.1007/s10802-011-9537-z), so RPGs can break this cycle by acting as a method for scaffolding social skills in a regulated, consequence-free space.
- Response 7: This is a great resource. We added it to build on this. This can be found on page 3, lines 146-148.
- “Social withdrawal and impairment increase the risk of depression in adulthood (Katz et al., 2011); however, developing social skills through RPGs in a structured, low-risk environment may help mitigate this risk.”
- Comment 8: Another addition I’d like to see is in the Discussion. On line 585, it’s mentioned that a limitation of the present study is that comparative value against other interventions hasn’t been tested. To guide future research, I would encourage you to make suggestions as to why and how RPGs might be preferable to other forms of socialising or CBT. Aside from autism, are there other people for whom this approach could work better than traditional therapy? For instance, I imagine it could be more appealing to people who like games, less intimidating to people who are sceptical or uncomfortable in traditional therapy, and may help with attrition as there is an added incentive for continuation.
- Response 8: These are great points. We expand on this in the revised future research paragraph. This is on page 14, lines 655-670.
- “Future research should explore the broader applicability of TTRPGs across diverse populations and therapeutic settings. Longitudinal studies could examine the lasting effects of role-playing on identity formation and mental health, while comparative research could clarify the unique contributions of TTRPGs relative to other therapeutic modalities, such as cognitive-behavioral therapy (CBT). Additionally, investigating the effectiveness of online role-playing games as a therapeutic tool is crucial, particularly as technology continues to transform social interaction and mental health care. Research from different cultural spheres and societies would also be valuable, as the rapid diversification of role-playing game formats presents new opportunities to assess their therapeutic potential across various contexts. Further inquiry should explore why and how TTRPGs might be preferable to other forms of socializing or structured interventions like CBT, particularly for individuals who may find traditional therapy intimidating or unappealing. Beyond autism, this approach may be especially beneficial for individuals who enjoy games, those skeptical of conventional therapy, and those at risk of early attrition in treatment due to a lack of intrinsic motivation. By identifying the populations for whom TTRPGs may be most effective, future studies can help refine and optimize their use as a therapeutic tool.”
- Comment 9: Lastly, there is a small typo in the DOI for reference 30.
- Response 9: Thank you for pointing this out. I corrected the DOI from “https://doi-org/10.1080/10538712.2012.722592” to “https://doi.org/10.1080/10538712.2012.722592” on line 740.